# Re: Data Poisoning Attacks Against Multimodal Encoders

## Abstract

Multimodal models, which leverage both visual and linguistic modalities, have gained increasing attention in recent years. However, these models are often trained on large-scale unlabeled datasets, which expose them to the risk of data poisoning attacks. An adversary can manipulate the training data to induce malicious behaviors in the model under certain conditions. Yang et al. (2023) recently conducted a study on the susceptibility of multimodal models to poisoning attacks. They introduced three types of poisoning attacks targeted at multimodal models, along with two potential defenses. In this work, we replicate all three attack strategies. However, we observed that the effectiveness of the attack depends on the poisoning rate in relation to the quantity of samples in the targeted class, a factor that can potentially reduce the efficiency of the attack. Additionally, we replicated the ablation study, verified the consistency of their claims, and provided further experimentation to test them. Regarding the proposed defenses, we reproduced them and explained a flaw in the first defense. Furthermore, we propose a more practical setting for the second defense. We provide the code for reproducing the experiments here: [link]

## 1 Introduction

The limitations of machine learning (ML) models that contain a single modality have become increasingly apparent (Radford et al., 2021). This has sparked a growing interest in multimodal models, which, unlike their unimodal counterparts, utilize multiple modalities, enabling them to achieve exceptional performance in tasks such as image classification (Radford et al., 2021) and image captioning (Laina et al., 2019; Mokady et al., 2021). However, like other machine learning models, these models are susceptible to various security and privacy issues, including inference attacks (Shokri et al., 2017; Zhou et al., 2022), adversarial manipulation (Ilyas et al., 2019; Xie et al., 2019), and data poisoning (Wang et al., 2022).

A recent study by Yang et al. (2023) centered on the execution of data poisoning attacks on multimodal models. This research targets both the linguistic and visual modalities, aiming to ascertain whether the linguistic modality is as susceptible to poisoning attacks as the visual modality, which was previously proven to be vulnerable by Carlini & Terzis (2022). Moreover, Yang et al. (2023) strive to determine which modality is more susceptible to these attacks. They introduced three types of poisoning attacks and studied the impact of altering the poisoning rates, dataset sizes, and image encoders. They also experimented with transferring the attack by poisoning one dataset and assessing the attack on another. Finally, they proposed two defense strategies to mitigate the effects of the attacks.

In this study, we analyze all three attacks that have been proposed. We evaluate the effectiveness of these attacks in the same settings in which they were originally proposed. Additionally, we determine that their success largely depends on the poisoning rate relative to the number of samples in the targeted class (poisoning rate relative to class size), contrary to what was suggested by Yang et al. (2023), who discussed the effect of poisoning rates relative to the total number of samples in the dataset. This factor can prove the attacks inefficient as they become much harder to perform and easier to detect. We then assess the transferability of the attack from one dataset to another, and further extend our investigation by testing the transferability using various poisoning rates. We explore the effects of changing dataset sizes, image encoder sizes, and balancing the datasets. We replicate both defenses, identify a shortcoming in the first defense, and propose a more practical variant of the second.

The remainder of this paper is structured as follows: Section 2 provides background information about the task, attacks, and defenses. Section 3 details the running setting and the useful resources provided in the original paper. This section also elucidates the experiments and presents the results. Finally, Section 4 concludes our study and discusses potential future work.

## 2 Background

This section aims to provide information about the specific task utilized for training the model, along with the strategies used for both poisoning and defenses.

### 2.1 Image Retrieval

This task involves the retrieval of images based on text queries. It is particularly designed for scenarios where there is a modality mismatch between the queries and the retrieval galleries. In other words, the queries are presented in one modality (text), while the retrieval galleries are in another modality (images) (Cao et al., 2022).

### 2.2 Attack Methodology

**Attack I: single target image.** This attack considers a simple scenario where the adversary aims to poison texts in one class (e.g.,"lamb on the grass") to a single image belonging to another class (e.g., aeroplane). This is done by adding samples with text captions and the target image. The attack is successful if the model utility is not affected while being able retrieve images from the target class using text captions from the class used for poisoning.

**Attack II: single target label.** In this attack, the adversary aims to map texts in one class (i.e., original class) to images in another class (i.e., target class). Note that here only one original class and one target class are selected. Unlike attack I, there are multiple images from the target class.

**Attack III: multiple target labels.** Attack III considers achieving multiple "single target label" poisoning attacks (Attack II) simultaneously, i.e., texts of multiple original classes are mapped to multiple target classes simultaneously. Attack III differs from attack II as it requires the model to learn multiple "mismatched" relationships, i.e., to "remember" multiple poisoned relationships, with a one-time injection of poisoned samples.

### 2.3 Defenses

**Pre-training defense.** This defense mechanism employs a pre-trained image encoder to compute the cosine distance between the image-caption pairs. A threshold for the maximum cosine distance is set and all samples exceeding this threshold are discarded.

**Post-training defense.** This defense strategy relies on the availability of a clean dataset that is similar to the poisoned dataset. This clean dataset is utilized to sanitize the model after it has been trained on the poisoned data, by further training it using this clean data.

## 3 Experiments

This section details the conducted experiments, presents the findings, and delves deeper into significant observations. We begin by executing all three attacks and comprehensively analyzing their impacts. Subsequently, we isolate and poison each modality independently, demonstrating the vulnerability of both. We then check the effect of poisoning rate relative to the entire dataset and further explore the effect of poisoning rate relative to the target class size which was not done previously. Moving forward, we evaluate the transferability of the attack by applying poisoning to one dataset and assessing its impact on another. We face a challenge in replicating the same results as those in the original paper because the poisoning rate was not reported. Subsequently, we demonstrate how to regulate its effectiveness as this was not reported.

We also experiment the effect of changing the dataset size and image encoder size. Further experimentation investigates attack feasibility on a balanced dataset. Lastly, we replicate both defense strategies, revealing a weakness in the first and proposing a more realistic alternative for the second. All these experiments were executed on a single Nvidia Quadro RTX 6000 GPU with 24GB of memory.

## 3.1 Materials provided in the original paper

The authors of the original study have made their experimental code[1] accessible to the public. However, we discovered that this code only facilitates the execution of two or three out of the nine experiments they actually conducted, but the provided code significantly eased our task. Upon examining the code, we identified a discrepancy: the authors computed poisoning rates relative to the number of samples in each class in their code, whereas in their paper, they referred to it relative to the total number of samples in the dataset. As a result, we report both poisoning ratios for each experiment we conduct in appendix A.

While this discrepancy might not be a major concern, as the poisoning targets the encoder using image-caption pairs rather than classes, it could potentially simplify the detection of the attack if the user decides to label the dataset using keywords from the captions. The authors used this labeling method for one of the used datasets. Therefore, if an adversary can exploit the poisoning of the encoder to match image-text pairs from different classes, the user could also inspect these classes as they might be of value. Consequently, if the class-relative poisoning ratio is high, it could lead to the detection of the attack.

## 3.2 Experimental Settings

We follow all the settings described in Yang et al. (2023), which we reiterate here for the convenience of the reader.

**Target Models.** We used only the CLIP model (Radford et al., 2021), using ViT-B/32 (Dosovitskiy et al., 2021) as the default image encoder and a transformer (Vaswani et al., 2017) with modifications in Radford et al. (2019) as the text encoder. The attacks are performed during fine-tuning of the pre-trained model, with a maximum text sequence length of 76. We employ an Adam optimizer with decoupled weight decay regularization, and a cosine scheduler for learning rate decay. The initial learning rate is $10^{-5}$ , with a weight decay rate of 0.2. The cosine scheduler has a minimum learning rate of $10^{-6}$ and a decay rate of 1.0. The model is fine-tuned for 10 epochs with a batch size of 128.

Table 1: Dataset statistics

| Dataset | # Pairs | # Images | # Labeled Images | # Classes |
|---|---|---|---|---|
| Flickr | 158,915 | 31,873 | - | - |
| PASCAL | 4,998 | 1,000 | 1,000 | 20 |
| COCO | 616,767 | 123,287 | 122,218 | 80 |
| VG | 540,378 | 108,077 | - | - |

**Dataset.** We employ the same four datasets as in the original paper, namely Flickr30k(Young et al., 2014), COCO(Chen et al., 2015), PASCAL(Rashtchian et al., 2010), and Visual Genome(Krishna et al., 2017). The statistics for these datasets are provided in Table 1. To address the limited size of the PASCAL dataset, a merged Flickr-PASCAL dataset is created. This combined dataset includes 29k samples from Flickr and 500 from PASCAL for training, with each sample comprising one image and five text captions which is the case for all datasets. The remaining 500 samples from PASCAL are used for testing. The authors provided the splits used for Flicker-PASCAL and COCO which we use in our experiments.

In **Attack I**, our goal is to poison *sheep*-labelled texts to a single target *aeroplane* image for *Flickr-PASCAL*, and *boat*-labelled texts to a single target *dog* image for *COCO*. For **Attack II**, we randomly select the samples to be poisoned. The poisoning goals for *Flickr-PASCAL* and *COCO* are *sheep2aeroplane* and *boat2dog*,

---

[1]Authors' Code: https://github.com/zqypku/mm$_p$oison/

respectively. The *Flickr-PASCAL* dataset is poisoned with 25 samples (125 pairs) for a poisoning rate of 0.08%, while the *COCO* dataset is poisoned with 284 samples (1,420 pairs) for a rate of approximately 0.24%.

In **Attack III**, we poison the model with two goals for each dataset: *sheep2aeroplane* and *sofa2bird* for *Flickr-PASCAL*, and *boat2dog* and *zebra2train* for *COCO*. We poison the training data of each dataset based on these goals with a one-time injection. The resulting poisoning rates for *Flickr-PASCAL* and *COCO* are 0.16% and 0.52%, respectively.

In evaluating poisoning attacks, we employ three metrics. The first, Hit@K, calculates the fraction of text samples for which the target images appear within the first K entities of the rank list for the image retrieval task. We consider Hit@1, Hit@5, and Hit@10, with larger values indicating a better rank list as more text samples can hit target images early. The second metric, MinRank, is the minimum rank of the target images in the rank list of all test images, with smaller values indicating a better rank list as target images can be seen earlier. Lastly, we use Cosine distance, a measure of similarity between two embeddings that ranges between 0 and 2, with values closer to 0 indicating similar embeddings.

We assess the effectiveness of the poisoning attack by computing the Hit@K and average MinRank across all test images, with success indicated by a higher Hit@K and lower MinRank. For the baseline, we randomly select an equivalent number of texts from the test data for target image retrieval (random2target). The model's utility is quantified by comparing the average Hit@K of the poisoned and clean models for both image and text retrieval across batches of images where the ground truth consists of (text, image) pairs. A closer Hit@K rate denotes a higher model utility.

### 3.3 Performing All Three Attacks

We start by executing all three attacks using the previously explained default settings. In the tables, **Baseline** refers to the results obtained when retrieving the target image class using random text. Conversely, **Ours** represents the results when retrieving the target image class using the corresponding target text class. These notations are used in alignment with the original paper. The original paper did not evaluate retrieving the target images using the clean model and only used the poisoned models for both baseline results by retrieving random2target. We find it more convenient to include this evaluation on the clean to show the actual performance of attacks over the clean model.

This experiment was conducted using the authors' code without any modifications. Despite the original paper stating that the target image is selected randomly, we opted to use the hardcoded images in the code for **Attack I**, ensuring a consistent execution of the attack. These images are depicted in figures 1 and 2.

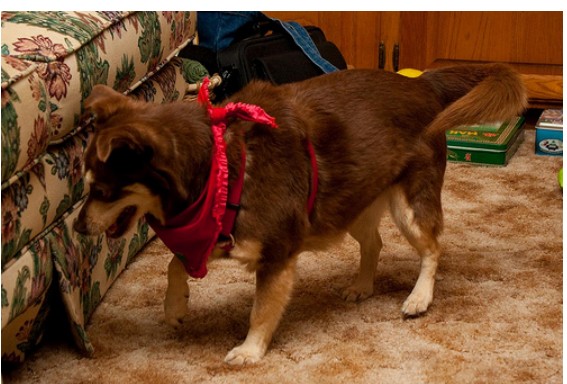

Figure 1: Target image for *COCO* used in Attack I, as hardcoded in the original paper's code.

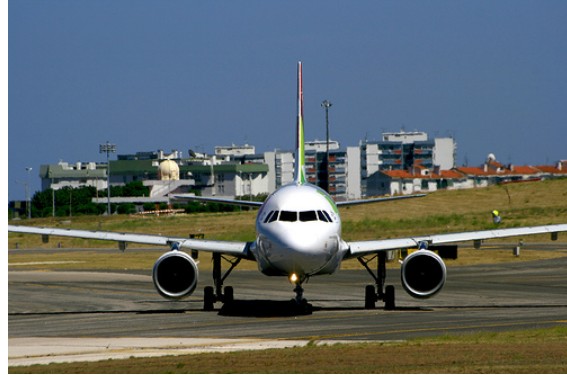

Figure 2: Target image for *Flickr-PASCAL* used in Attack I, as hardcoded in the original paper's code.

As depicted in Table 2, the utility variation between the poisoned and clean models is barely noticeable. The relatively small poisoning ratio in relation to the entire dataset, had almost no effect on the utility. In

Table 2: Comparison of Utility (Hit@10) Between Clean and Poisoned Models: The differences are minimal, consistent with the findings in the original paper.

| Dataset | Task | Clean | Attack I | Attack II | Attack III |
|---|---|---|---|---|---|
| Flickr-PASCAL | TR | 0.986 | 0.980 | 0.982 | 0.978 |
| | IR | 0.968 | 0.969 | 0.964 | 0.966 |
| COCO | TR | 0.927 | 0.933 | 0.935 | 0.936 |
| | IR | 0.864 | 0.865 | 0.864 | 0.866 |

fact, it can even increase, as demonstrated in the *COCO* case. These findings align with those reported in the original paper.

Table 3: Performance Outcomes of Attack I on *Flickr-PASCAL* and *COCO* Datasets. The clean models are added as an extra comparison over the original paper. The performance on the *Flickr-PASCAL* dataset, specifically in terms of Hit@1 and Hit@5, is lower than the original paper's results.

| Dataset | Method | Hit@1 | Hit@5 | Hit@10 | MinRank |
|---|---|---|---|---|---|
| Flickr-PASCAL (Clean) | Baseline | 0.016 | 0.032 | 0.072 | 77.176 |
| | Ours | 0.000 | 0.000 | 0.016 | 69.264 |
| Flickr-PASCAL (Poisoned) | Baseline | 0.016 | 0.056 | 0.144 | 45.28 |
| | Ours | 0.024 | 0.488 | 0.864 | 5.672 |
| COCO (Clean) | Baseline | 0.016 | 0.052 | 0.120 | 155.52 |
| | Ours | 0.004 | 0.052 | 0.176 | 43.728 |
| COCO (Poisoned) | Baseline | 0.016 | 0.092 | 0.140 | 96.484 |
| | Ours | 0.032 | 0.456 | 0.764 | 9.576 |

Table 3 presents the results of **Attack I** on both the *Flickr-PASCAL* and *COCO* datasets, highlighting the attack's effectiveness over both the baseline (random2target) and the clean model. The *COCO* dataset results align closely with those reported in the original paper. However, on *Flickr-PASCAL*, we obtained Hit@1, Hit@5, and Hit@10 values of 0.024, 0.488, and 0.864 respectively, which are lower than the original paper's reported values of 0.320, 0.928, and 0.968. This discrepancy is surprising given that we used the same code, hardcoded target image, and settings as Yang et al. (2023), including the provided train and test splits for this experiment. Despite numerous trials on the *Flickr-PASCAL* dataset, the output consistently deviated from the original paper's results.

The performance of **Attack II**, as shown in Table 4, closely mirrors the original paper's results, with a minor decrease observed on *Flickr-PASCAL*. For the *COCO* dataset, slightly improved results were achieved on Hit@5 and Hit@10. The MinRank values indicate the minor discrepancies between our results and those of the original paper.

Table 5 displays the results of **Attack III**. As noted in the original paper, the outcomes of poisoning the second goal differ from those of the first goal. This discrepancy is not observed in the *COCO* dataset, where similar results are obtained for both goals. The authors did not elaborate on why **Attack III** performs differently on *Flickr-PASCAL* compared to *COCO*. In the *COCO* case, doubling the poisoning rate nearly doubles the attack performance, split across the two classes. This phenomenon was not observed in the *Flickr-PASCAL* case. Repeated experiments yielded consistent results. It is also worth noting that for all three attacks, the clean model boat2dog evaluation had better results than the baseline (random2target) that was used in the original paper.

Table 4: Performance Outcomes of Attack II on *Flickr-PASCAL* and *COCO* Datasets. Our results closely align with those of the original paper, with a slight decrease on *Flickr-PASCAL* and a slight increase on *COCO*.

| Dataset | Method | Hit@1 | Hit@5 | Hit@10 | MinRank |
|---|---|---|---|---|---|
| Flickr-PASCAL (Clean) | Baseline | 0.016 | 0.032 | 0.072 | 77.176 |
| | Ours | 0.000 | 0.000 | 0.016 | 69.264 |
| Flickr-PASCAL (Poisoned) | Baseline | 0.024 | 0.088 | 0.216 | 49.248 |
| | Ours | 0.152 | 0.768 | 0.920 | 3.6 |
| COCO (Clean) | Baseline | 0.016 | 0.052 | 0.120 | 155.52 |
| | Ours | 0.004 | 0.052 | 0.176 | 43.728 |
| COCO (Poisoned) | Baseline | 0.020 | 0.064 | 0.108 | 134.108 |
| | Ours | 0.024 | 0.280 | 0.544 | 14.912 |

Table 5: Results of Attack III on *Flickr-PASCAL* and *COCO*. In *COCO*, doubling the poisoning rate nearly doubles the attack performance, a phenomenon not observed in *Flickr-PASCAL*.

| Dataset | Method | Hit@1 | Hit@5 | Hit@10 | MinRank |
|---|---|---|---|---|---|
| Flickr-PASCAL (Clean) | Baseline-1 | 0.016 | 0.032 | 0.072 | 77.176 |
| | Ours-1 | 0.000 | 0.000 | 0.016 | 69.264 |
| | Baseline-2 | 0.040 | 0.080 | 0.128 | 40.84 |
| | Ours-2 | 0.000 | 0.008 | 0.016 | 81.104 |
| Flickr-PASCAL (Poisoned) | Baseline-1 | 0.016 | 0.080 | 0.168 | 47.976 |
| | Ours-1 | 0.096 | 0.680 | 0.904 | 4.248 |
| | Baseline-2 | 0.056 | 0.120 | 0.248 | 28.864 |
| | Ours-2 | 0.016 | 0.296 | 0.616 | 12.328 |
| COCO (Clean) | Baseline-1 | 0.016 | 0.052 | 0.120 | 155.52 |
| | Ours-1 | 0.004 | 0.052 | 0.176 | 43.728 |
| | Baseline-2 | 0.016 | 0.036 | 0.052 | 373.672 |
| | Ours-2 | 0.000 | 0.000 | 0.000 | 192.46 |
| COCO (Poisoned) | Baseline-1 | 0.016 | 0.068 | 0.108 | 143.964 |
| | Ours-1 | 0.012 | 0.172 | 0.452 | 16.128 |
| | Baseline-2 | 0.016 | 0.040 | 0.052 | 251.992 |
| | Ours-2 | 0.008 | 0.100 | 0.424 | 13.96 |

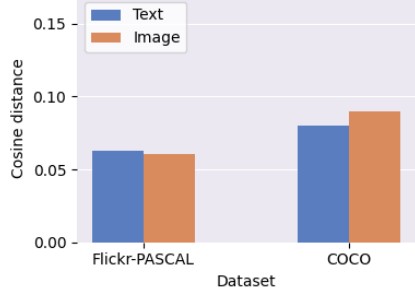

Figure 3: The cosine distance for both datasets using encoders from Attack II.

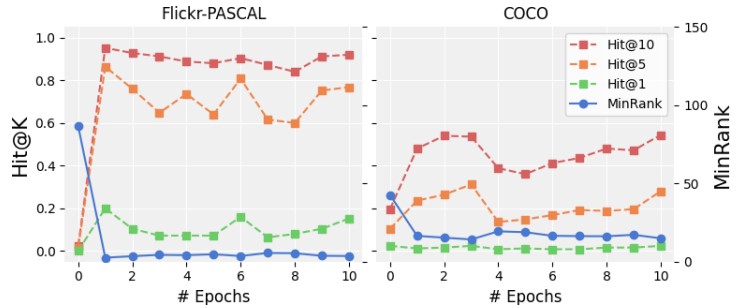

Figure 4: The impact of Attack II emerges in the initial epochs and stabilizes as the model continues to learn.

Figure 3 illustrates the cosine distance between the clean and poisoned encoders for both datasets. The encoders from **Attack II** were used, as they most closely matched the original paper's results. However, our results differed. The original paper's cosine distance results contradicted their appendix tables, which contained the actual cosine distance values. We therefore conclude that cosine distances cannot be used as a reliable measure of attack effectiveness on each modality. Figure 4 demonstrates that the impact of **Attack II** emerges in the initial epochs and stabilizes as the model continues to learn.

### 3.4 Poisoning Each Modality

We investigate the impact of freezing the weights of one encoder while training the other. The notation $M_p$ denotes that both encoders are trainable. $M_p^i$ signifies that the image encoder is trainable with the text encoder frozen, while $M_p^t$ indicates that the text encoder is trainable with the image encoder frozen. $M_p^0$ refers to the use of a pre-trained model without any fine-tuning.

Table 6: Impact of Freezing Encoder Weights on Attack Performance. The results align with the original paper, except for the *Flickr-PASCAL* dataset where the MinRank of the image encoder was not consistently lower than that of the text encoder. These findings confirm the vulnerability of both modalities to poisoning, but definitive assumptions about the poisoning effect on each modality require further investigation.

| Dataset | Model | Hit@1 | Hit@5 | Hit@10 | Hit@20 | Hit@30 | Hit@50 | MinRank |
|---|---|---|---|---|---|---|---|---|
| Flickr-PASCAL | $M_p$ | 0.152 | 0.768 | 0.920 | 0.984 | 0.992 | 1.000 | 3.6 |
| | $M_p^i$ | 0.056 | 0.696 | 0.928 | 0.976 | 0.992 | 1.000 | 4.272 |
| | $M_p^t$ | 0.216 | 0.712 | 0.896 | 0.992 | 0.992 | 1.000 | 3.704 |
| | $M_p^0$ | 0.000 | 0.016 | 0.024 | 0.040 | 0.128 | 0.224 | 86.568 |
| COCO | $M_p$ | 0.024 | 0.280 | 0.544 | 0.832 | 0.876 | 0.920 | 14.912 |
| | $M_p^i$ | 0.008 | 0.284 | 0.416 | 0.748 | 0.852 | 0.940 | 16.728 |
| | $M_p^t$ | 0.056 | 0.284 | 0.452 | 0.728 | 0.820 | 0.900 | 21.028 |
| | $M_p^0$ | 0.024 | 0.104 | 0.196 | 0.380 | 0.544 | 0.688 | 42.636 |

The results presented in Table 6 align with those reported by Yang et al. (2023). However, the assumption that the MinRank of the image encoder will consistently be lower than that of the text encoder, did not hold true for the *Flickr-PASCAL* dataset. Our MinRank was 4.272, compared to 3.016 in the original paper. Given the close proximity of the results, it is challenging to make definitive assumptions about the poisoning effect on each modality and its potential benefits in each case, as this would necessitate more rigorous proofs and extensive experiments. Nevertheless, these results confirm that both modalities are vulnerable to poisoning.

### 3.5 Effect of Poisoning Rate

We investigate the effect of different poisoning rates on the success of the attack. In line with Yang et al. (2023), we utilize **Attack II** for this investigation, using the same poisoning ratios reported relative to the total number of samples in the dataset. The hyperparameters are kept consistent with those used in previous experiments, the poisoning rate is the only variable that we adjust. Our findings align with those of the original paper, with the exception of the 0% poisoning rate. The authors stated that they utilized a clean model for this rate. However, it appears that the results from their poisoned baselines were used instead, as the graph displays numbers similar to those reported for their baseline.

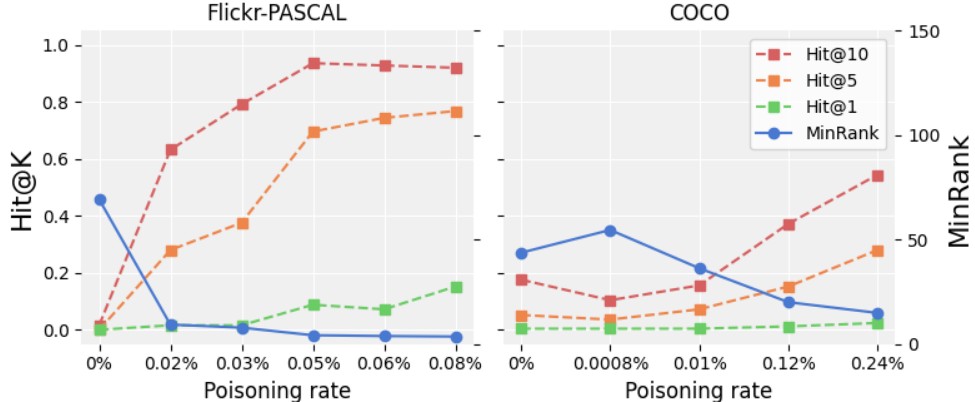

Figure 5: The impact of varying poisoning rates reveals an enhancement in performance on *Flickr-PASCAL* with an increase in the poisoning rate, while *COCO* only shows improvement after reaching a 0.12% rate, even though it had higher rates. Our findings align with the original paper, with the exception of the 0% poisoning rate.

As illustrated in Figure 5, the performance on the *Flickr-PASCAL* dataset enhances with an increase in the poisoning rate, with a mere 0.05% proving sufficient to achieve peak performance. In contrast, the *COCO* dataset exhibits negligible performance improvement until the poisoning ratio reaches 0.12%, beyond which the performance begins to improve. Despite the higher poisoning rates for *COCO* compared to *Flickr-PASCAL*, the poisoning outcomes consistently favor the latter.

### 3.6 Poisoning Relative to Class Size

While this behaviour was not explained by Yang et al. (2023), we attribute it to the influence of the poisoning ratio in relation to the class size. If the multimodal model is fine-tuned with a sufficient number of clean samples from the poisoned class, and these samples exceed a certain threshold, the impact of the attack becomes negligible. To validate this, we conducted an additional experiment where we varied the poisoning rates relative to the size of the poisoned class for the *Flickr-PASCAL* dataset, maintaining the same poisoning goal of *sheep2aeroplane*. It is noteworthy that a poisoning rate of **0.08%** corresponds to **100%** of the samples in the poisoned class. All poisoning ratios and number of samples are mention in appendix A

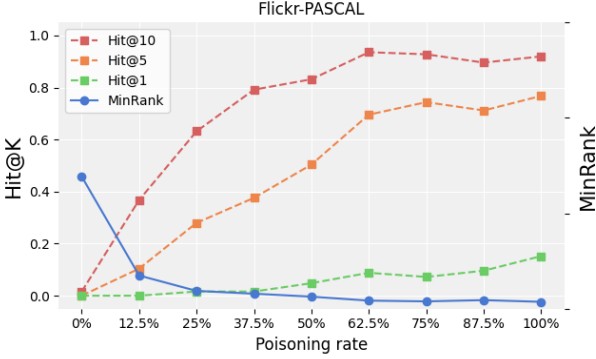

Figure 6: The influence of poisoning ratios relative to the class size demonstrates a more consistent result.

Figure 6 illustrates the enhancement in poisoning performance as the poisoning ratio relative to the class size increases, reaching a saturation point beyond 50% of the class size. This implies that to achieve the maximum

attack performance, poisoning is required for 50% of the class samples. However, our new experiment modifies the poisoning rates relative to the entire dataset. Therefore, we conducted an additional experiment leveraging the class imbalance in *COCO*, altering the poisoning goal from *boat2dog* to *scissors2toothbrush*. The **0.24%** poisoning rate for the first goal corresponds to approximately **20%** of the class samples, while it represents around **80%** of the class samples for the second poisoned class. This approach enables us to maintain a constant **0.24%** relative dataset ratio while solely adjusting the poisoning rate relative to the class.

Table 7: Variation in performance is observed in the *COCO* dataset, despite maintaining a constant poisoning ratio of 0.24%. This variation is evident when the poisoning goal is altered from 'boat2dog' to 'scissors2toothbrush'. Notably, even with the poisoning ratio remaining constant relative to the dataset, the attack performance differs for each class.

| Dataset | Method | Hit@1 | Hit@5 | Hit@10 | MinRank |
|---|---|---|---|---|---|
| COCO (Clean) | Baseline | 0.016 | 0.052 | 0.120 | 155.52 |
| | Ours | 0.004 | 0.052 | 0.176 | 43.728 |
| boat2dog (Poisoned 19.9%) | Baseline | 0.020 | 0.064 | 0.108 | 134.108 |
| | Ours | 0.024 | 0.280 | 0.544 | 14.912 |
| scissors2toothbrush (Poisoned 79.7%) | Baseline | 0.012 | 0.040 | 0.072 | 175.904 |
| | Ours | 0.156 | 0.596 | 0.760 | 22.208 |

Table 7 illustrates the performance variation despite the constancy of the poisoning ratio relative to the dataset. It is noteworthy that the attack performance does not match the levels achieved for the *Flickr-PASCAL* dataset. This aspect remains unexplored and may necessitate a more comprehensive study to understand the attack's impact on the dataset and the fine-tuning of the utilized model.

The preceding results suggest that these attacks may not be as effective as initially proposed by the authors. The underlying assumption was that an adversary could poison the network and achieve a small poisoning ratio relative to the dataset, which would yield significant attack performance. However, given that the attack's effectiveness depends on the relative class size, this may not always be the case, as the adversary cannot guarantee the attack performance. Even though the attack targets the entire dataset, if poisoning from A to B is of interest to the adversary, the user could implement some basic checks to prevent the attack from succeeding or add enough clean sample to limit the attack's performance.

### 3.7 Attack Transferability

In order to evaluate the transferability of the attack across different datasets, we adopted the method proposed by Yang et al. (2023) to poison the *Visual Genome* dataset with the goal *sheep2aeroplane* and evaluate the impact of this poisoning on the *Flickr-PASCAL* test set. As indicated in Table 8, maintaining a consistent poisoning ratio relative to the dataset (0.08%), but with a class-relative poisoning ratio of 6.7%, does not replicate the attack performance reported in the original paper, where the specific poisoning ratio for this attack on *Visual Genome* is not provided. However, when the poisoning ratio relative to the class is set to 100%, mirroring the approach used in **Attack II** on *Flickr-PASCAL*, we were able to match the performance reported in the original paper, as shown in Table 8.

Given the original paper's lack of clarity on how to effectively transfer the poisoning attack across different datasets, we decided to further investigate this phenomenon. Our experiment involved the use of a clean model, the maximum possible poisoning of 1.18%, and four other intermediate poisoning ratios (0.08%, 0.1%, 0.5%, 0.8%). These poisoning ratios were also examined relative to the class size. The objective was to ascertain the degree of transferable poisoning and its dependencies. As illustrated in Figure 7, the results indicate that the performance of the attack's transferability is primarily determined by the poisoning ratio in relation to the class size.

Table 8: Evaluation of Attack II's transferability performance using the *Visual Genome* dataset and tested on the *Flickr-PASCAL* dataset. The performance of Attack II is significantly lower when the poisoning ratio relative to the dataset size is the same as that used for *Flickr-PASCAL*, it matches the expected performance when the poisoning is executed using the ratio relative to the class size.

| Dataset | Method | Hit@1 | Hit@5 | Hit@10 | MinRank |
|---|---|---|---|---|---|
| VG (Clean) | Baseline | 0.016 | 0.040 | 0.120 | 66.896 |
| | Ours | 0.000 | 0.000 | 0.016 | 68.128 |
| VG (Poisoned 0.08%) | Baseline | 0.008 | 0.040 | 0.080 | 70.936 |
| | Ours | 0.000 | 0.000 | 0.152 | 19.52 |
| VG (Poisoned 1.18%) | Baseline | 0.024 | 0.136 | 0.176 | 69.36 |
| | Ours | 0.344 | 0.864 | 0.928 | 3.64 |

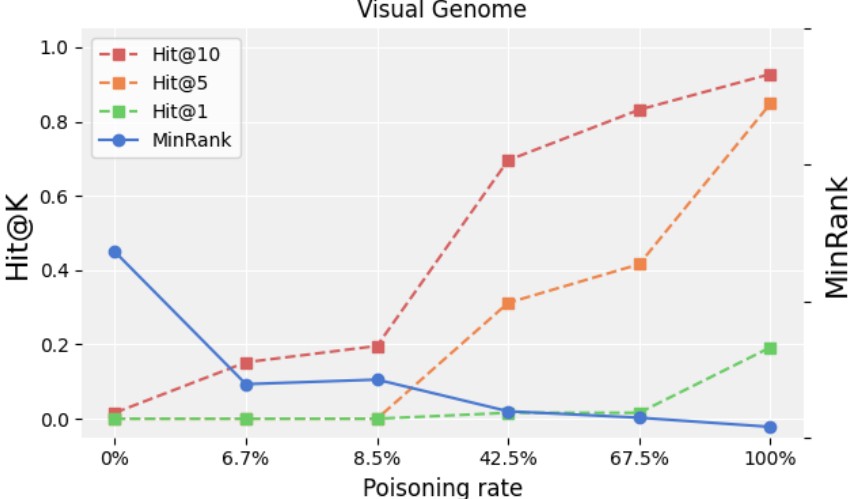

Figure 7: Performance of the transferability of the poisoning attack across different datasets, specifically from VG to Flickr-PASCAL. The poisoning ratios, expressed relative to the class size, correspond to 0.0%, 0.08%, 0.1%, 0.5%, 0.8%, and 1.18% of the dataset size. The performance is primarily influenced by the poisoning ratio relative to the class size.

## 3.8 Dataset Size

To test the effect of dataset size, the *COCO* dataset is partitioned into two subsets: *COCO-S*, representing 25% of the original dataset size, and *COCO-M*, constituting 50% of the original dataset size. The splitting process is stratified to ensure the class ratio remains consistent across all three datasets. Subsequently, **Attack II** is executed on each of the three datasets, maintaining a poisoning ratio of 0.24%, as in previous experiments. As depicted in Figure 8, the attack performance appears to be independent of the dataset size, with nearly identical performance observed across all three splits.

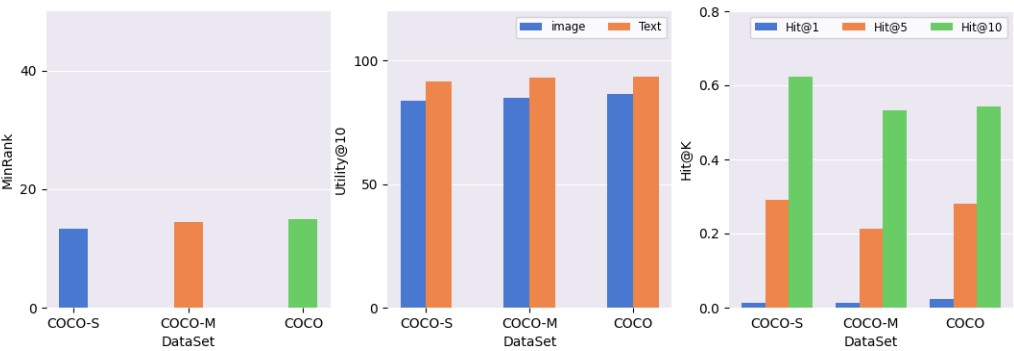

Figure 8: Evaluation of **Attack II** performance across varying dataset sizes. The *COCO* dataset is partitioned into two subsets: *COCO-S* (25% of *COCO*) and *COCO-M* (50% of *COCO*). Despite the change in dataset size, the attack performance remains consistent, indicating its independence from dataset size.

### 3.9 Changing Image Encoder

In order to examine the effect of altering the image encoder, we investigated the impact of utilizing various image encoders on the effectiveness of **Attack II**. In accordance with the settings of the original paper, we employed ViT-B/32, ViT-B/16, and ViT-L/14. The experimental setup remained consistent with previous settings, with the sole alteration being the substitution of the image encoder. As depicted in Figure 9, the attack maintains its effectiveness across different image encoder sizes for both datasets, suggesting that the success of the attack is not dependent on a specific image encoder size.

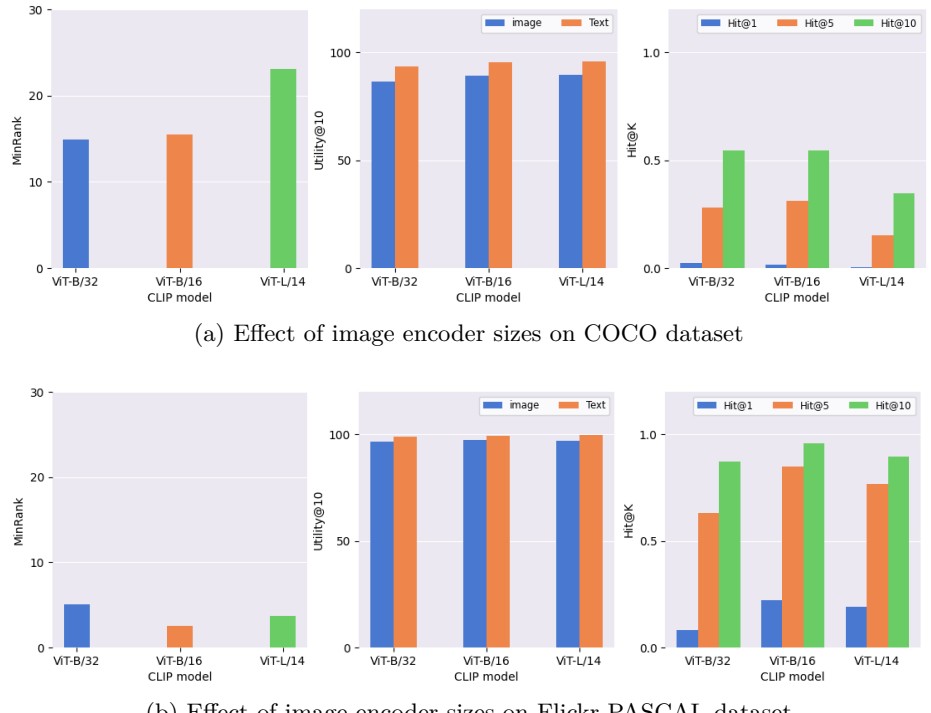

(a) Effect of image encoder sizes on COCO dataset

(b) Effect of image encoder sizes on Flickr-PASCAL dataset

Figure 9: Impact of varying image encoder sizes on different datasets. The effectiveness of **Attack II** is not significantly influenced by changes in the image encoder size. ViT-B/32 is considered as the baseline, as it has been used in all previous experiments.

### 3.10 Attacking Balanced Datasets

Until now, all conducted experiments involved five text captions for each image. We follow the same setup as Yang et al. (2023), selecting a random caption for each image to establish a balanced dataset. The same settings as previous experiments were employed for this test. However, the authors did not reveal the splits used for this experiment, which could potentially explain the differences in the results.

Table 9: Performance of Attack II on balanced datasets. **Attack II** performance improves on balanced *COCO* while slightly decreases on *Flickr-PASCAL*. These results are slightly below what was reported in the original paper.

| Dataset | Hit@1 | Hit@5 | Hit@10 | MinRank |
|---|---|---|---|---|
| Flickr-PASCAL-b | 0.160 | 0.680 | 0.840 | 4.68 |
| COCO-b | 0.020 | 0.340 | 0.610 | 14.78 |

Table 10: Cosine distance between clean and poisoned encoders on balanced datasets. Huge variance between these results and the results reported in the paper.

| Dataset | Text | Image |
|---|---|---|
| Flickr-PASCAL-b | 0.016 | 0.021 |
| COCO-b | 0.038 | 0.048 |

The results in Table 9 indicate that the attack remains effective on a balanced dataset. However, the Hit@5 and Hit@10 results for *Flickr-PASCAL* are lower than the reported values of 0.848 and 0.944, respectively, in the original paper. Similarly, the Hit@10 result for *COCO* is slightly lower, with a reported increase of almost 0.2 over the unbalanced dataset, an effect that was not attended in the original paper. Given the close performance results between the balanced and unbalanced datasets on our end, it is challenging to understand the reasons for such an improvement on *COCO* dataset in the original paper.

Moreover, Table 10 shows that the cosine distances for the balanced dataset are smaller than those for the unbalanced dataset, which deviates from the findings in the original paper. This observation reinforces our previous concern about the reliability of cosine distance as a metric for assessing the impact of poisoning.

### 3.11 Pre-training Defense

We test the first proposed defense. The defense mechanism operates by computing the cosine distances between pairs using a pre-trained ViT-B/16 model. Pairs with a cosine distance exceeding a specified threshold, in this case, **0.8**, are subsequently eliminated. Upon implementing this defense mechanism, we successfully detected 118 out of the 125 poisoned pairs. These pairs were then excluded from the training data, which allowed us to achieve the results presented in Table 11.

Table 11: Performance of pre-training defense against Attack II. The performance of the attack drops dramatically as 118 of the 125 poisoned pairs were dropped.

| Method | Hit@1 | Hit@5 | Hit@10 | MinRank |
|---|---|---|---|---|
| Attack II | 0.152 | 0.768 | 0.920 | 3.6 |
| Defense | 0.000 | 0.008 | 0.008 | 40.752 |
| Clean | 0.000 | 0.000 | 0.016 | 69.264 |

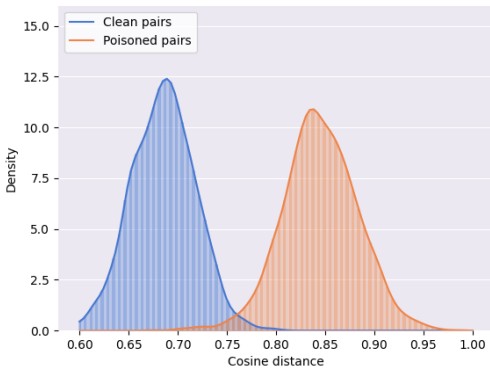

Figure 10: Probability density of cosine distances between clean/poisoned pairs in Flickr-PASCAL where all pairs are poisoned and it is clear where to set a threshold.

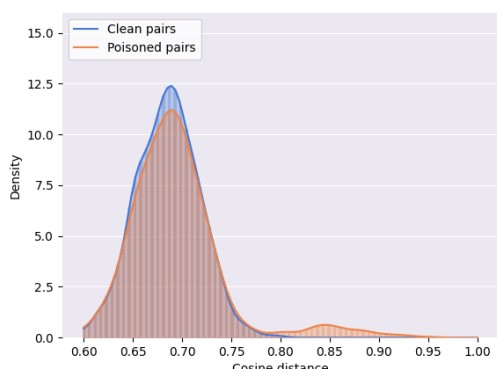

Figure 11: Probability density of cosine distances between clean/poisoned pairs in Flickr-PASCAL where only the pairs used for previous attacks are poisoned where it is not clear where to set a threshold.

However, the determination of an appropriate threshold proved to be a challenge, as illustrated by Figure 10 and Figure 11. In these figures, it is evident that we need to poison the entire dataset to ascertain the appropriate threshold. In real-world scenarios, especially with large datasets, the probability of detecting such anomalies can be challenging, and might depend on the poisoned classes.

### 3.12 Post-training Defense

This defense involves further fine-tuning the poisoned model using clean data, effectively sterilizing the model. We apply this defense to the models poisoned in the first experiment on both datasets using **Attack II**. The fine-tuning process is conducted using the clean *Visual Genome* dataset.

Table 12: Effectiveness of post-training defense. We observed a significant decrease in the utility for the *COCO* dataset, this deviates from the findings in the original paper.

| Dataset | Hit@10 (TR) | Hit@10 (IR) |
|---|---|---|
| Flickr-PASCAL | 0.962 | 0.919 |
| COCO | 0.823 | 0.693 |

Table 13: Impact of learning rate (LR) on defense. This is the first instance of altering the learning rate, as all previous experiments utilized the same rate. The defense appears to perform optimally at a learning rate of $10^{-4}$, which is consistent with the findings in the original paper.

| Method | LR | Hit@1 | Hit@5 | Hit@10 | MinRank |
|---|---|---|---|---|---|
| Attack II | - | 0.152 | 0.768 | 0.920 | 3.6 |
| Defense | $10^{-3}$ | 0.016 | 0.080 | 0.120 | 50.184 |
|  | $10^{-4}$ | 0.000 | 0.000 | 0.000 | 78.096 |
|  | $10^{-5}$ | 0.000 | 0.000 | 0.008 | 52.816 |

As demonstrated in Table 12, the utility on the same test set remains unaffected after further fine-tuning the model for 5 epochs in the case of the *Flickr-PASCAL* dataset. However, a noticeable impact on utility

Table 14: Impact of learning rate (LR) on utility. The utility experiences a significant decrease for both $10^{-4}$ and $10^{-3}$. This observation was not reported in the original paper, making the defense results for these learning rates insignificant.

| Method | LR | Hit@10 (TR) | Hit@10 (IR) |
|--------|-----|-------------|-------------|
| Attack II | - | 0.982 | 0.964 |
| Defense | $10^{-3}$ | 0.034 | 0.032 |
| | $10^{-4}$ | 0.634 | 0.609 |
| | $10^{-5}$ | 0.962 | 0.919 |

was observed for the *COCO* dataset. These results differ from the 0.975 (TR) and 0.945 (IR) reported in the original paper. This reported text retrieval (TR) and image retrieval utility (IR) is even higher than the reported 0.911 and 0.836 obtained using the clean model in their first experiment.

Table 13 presents the performance of this defense in relation to different learning rates. This marks the first instance of the authors experimenting with various learning rates, despite the consistent performance of $10^{-5}$, which was employed in all previous experiments. Upon examining the utility of other learning rates in Table 14, it was found that they tend to degrade performance. Consequently, these learning rates are deemed unsuitable, despite the potential for improved defense performance. This could explain the observed drop in utility for the *COCO* dataset, suggesting that updates done using *VG* may result in the loss of information from the initial fine-tuning. Figure 12 further illustrates that the defense can take effect in as little as one epoch.

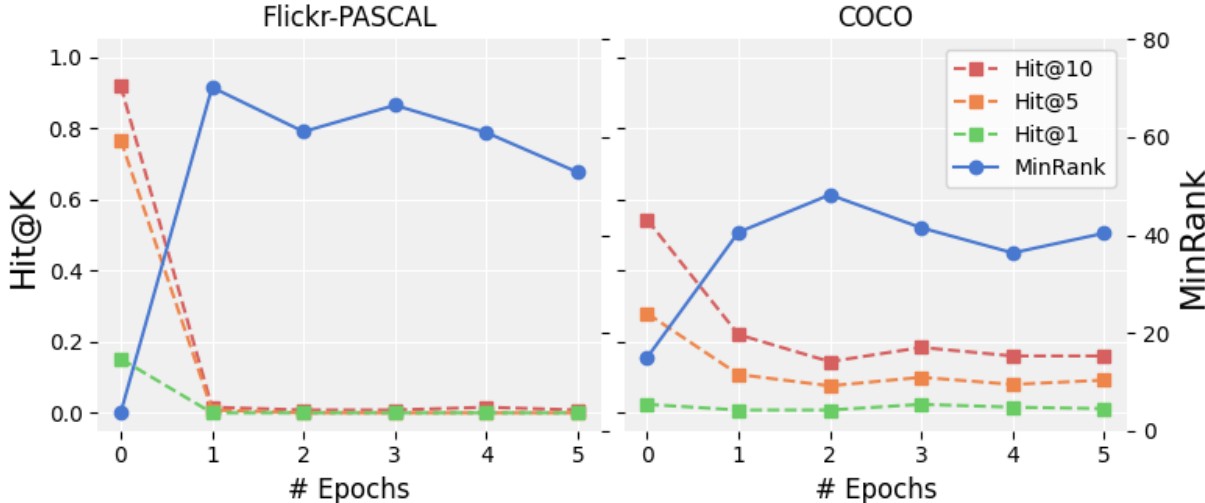

Figure 12: Effect of learning epochs on post training defense. One epoch is enough to observe the performance of the defense.

### 3.13 Enhanced Post-training Defense

The methodology proposed by Yang et al. (2023) raises a concern regarding the size of the clean dataset utilized in their defense mechanism. Specifically, with the *Flickr-PASCAL* dataset, the clean dataset surpasses the poisoned one in size, a situation that might not mirror actual circumstances. To rectify this, we introduce a modified version of the defense. Initially, we train the model on a large poisoned dataset, followed by further fine-tuning with a smaller clean dataset. This strategy is more practical as it is easier to inspect smaller datasets for potential poisoning.

We poison the model using **Attack II** on the *Visual Genome* dataset then further fine-tune it using the clean *Flickr-PASCAL* dataset to assess the performance of the defense. Subsequently, we excluded the *Flickr30k* dataset and focused solely on the 2500 clean pairs in the *PASCAL* dataset, thereby approximating a real-world scenario as closely as possible, as acquiring a clean dataset of 2500 pairs can be achieved.

Table 15: Post-train defence using smaller datasets. The defense performance using small datasets is almost as good as using the large clean dataset.

| Dataset | Method | Hit@1 | Hit@5 | Hit@10 | MinRank |
|---|---|---|---|---|---|
| VG (Poisoned 1.18%) | Baseline | 0.024 | 0.136 | 0.176 | 69.36 |
| | Ours | 0.344 | 0.864 | 0.928 | 3.64 |
| VG (Flickr-PASCAL) | Baseline | 0.040 | 0.096 | 0.152 | 70.304 |
| | Ours | 0.000 | 0.032 | 0.088 | 23.792 |
| VG (PASCAL) | Baseline | 0.040 | 0.080 | 0.096 | 89.28 |
| | Ours | 0.000 | 0.008 | 0.120 | 25.536 |

As shown in Table 15, the small clean dataset effectively neutralized the poisoning effect in both scenarios. The performance achieved using the *PASCAL* dataset was very close to that of the *Flickr-PASCAL* dataset. It's noteworthy that the model poisoned on *VG* datasets used 1275 samples, while the one poisoned on *Flickr-PASCAL* used only 25 samples. Despite this, the performance of the small dataset (25 clean samples after 1275 poisoned) is nearly identical to that of the large dataset used for defense (1275 clean samples after 25 poisoned).

Table 16: Utility of the Three Settings of post-train defense. The enhanced defense shows better utility values than the one proposed in the original paper.

| Setting | Text | Image |
|---|---|---|
| VG | 0.962 | 0.919 |
| Flickr-PASCAL | 0.976 | 0.953 |
| PASCAL | 0.980 | 0.958 |

As shown in Table 16, the utility increased by employing this approach, likely due to the more sophisticated gradient updates towards fewer classes, which are similar to what exists in the test set.

## 4 Conclusion

In this study, we successfully replicated Attack II and III. However, for Attack I, we achieved Hit@1, Hit@5, and Hit@10 values of 0.024, 0.488, and 0.864 respectively. These figures are lower than the original paper's reported values of 0.320, 0.928, and 0.968, despite using the code provided by Yang et al. (2023) and the hardcoded images for this attack. We observed that the effects of the attacks appear during the initial epochs and stabilize as the model continues training.

When testing each modality, we found that both are susceptible to a poisoning attack. For the *Flickr-PASCAL* dataset, we obtained a MinRank of 4.272, compared to 3.016 in the original paper. Although these two figures are close, it contradicts the original paper's claim that the MinRank of the image encoder is always less than the text encoder. We note that understanding how each modality is affected by poisoning and determining which is more vulnerable requires more rigorous proof and further investigation.

We then tested the effect of poisoning rates relative to dataset size and obtained similar results to what was reported in the original paper. We added an extra experiment to investigate the poisoning rates relative to the class size. We found that the primary factor for the attack's success is having a large poisoning rate

relative to the class size. This finding suggests that the attack is not as efficient as proposed, as it would be challenging to guarantee such a poisoning rate. It would also be much easier to detect the poisoning as our experiment shows that a poisoning rate of around 50% or more relative to class size can be required to make the attack efficient.

We tested the transferability of Attack II from one dataset to another and found that the authors did not use the 0.08% poisoning rate relative to the dataset size but actually 1.18%. We added another experiment to show that transferability also depends on the poisoning rate relative to the class size. We experimented using different dataset sizes and image encoders and obtained similar results to those published in the original paper.

We investigated balancing the datasets and obtained lower results than what the original paper reported. We achieved 0.610 and 0.860 for Hit@10 on *COCO* and *Flickr-PASCAL* respectively, while the original paper reported 0.712 and 0.944. We obtained an extra 0.2 on *COCO* compared to the unbalanced dataset, but no more details were reported for further investigation.

Finally, we implemented both proposed defenses, demonstrating the difficulty of setting a threshold for the first and proposing a more practical variant using smaller datasets for the second. This variant showed similar performance and better utility. However, while replicating the second defense, we failed to achieve the same utility on the *COCO* dataset as we obtained 0.823 (TR) and 0.639 (IR), while the paper reported 0.975 (TR) and 0.945 (IR), even higher than what they reported using their clean model.

For future work, there are several avenues for exploration, including studying the impact of each modality, experimenting with diverse multimodal models to assess attack effects, and investigating dataset characteristics and distributions in relation to the attacks.

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

# A    Dataset Details and Poisoning Rates

In this section, we provide additional insights into the datasets and their associated poisoning rates. We leveraged the splits published by the authors for both the *COCO* and *Flickr-PASCAL* datasets. However, for the balanced dataset experiment, where splits were not publicly available, we randomly selected one caption per image to create balanced datasets. The poisoning rates, relative to both dataset size and class size, are summarized in Tables 17 and 18 for both datasets.

Table 17: COCO Poisoning rates

| Goal | Class Samples | Poisoned Samples | Class poisoning rate | Dataset poisoning rate |
|---|---|---|---|---|
| boat2dog | 1422 | 284 | 19.9% | 0.24% |
| | | 142 | 9.98% | 0.12% |
| | | 12 | 0.84% | 0.01% |
| | | 1 | 0.07% | 0.0008% |
| zebra2train | 1700 | 284 | 16.7% | 0.24% |
| scissors2toothbrush | 356 | 284 | 79.7% | 0.24% |

Table 18: Flickr-PASCAL Poisoning rates

| Goal | Class Samples | Poisoned Samples | Class poisoning rate | Dataset poisoning rate |
|---|---|---|---|---|
| sheep2aeroplane | 25 | 25 | 100% | 0.08% |
| | | 21 | 87.5% | 0.07% |
| | | 18 | 75% | 0.06% |
| | | 15 | 62.5% | 0.05% |
| | | 12 | 50% | 0.04% |
| | | 9 | 37.5% | 0.03% |
| | | 6 | 25% | 0.02% |
| | | 3 | 12.5% | 0.01% |
| sofa2bird | 25 | 25 | 100% | 0.08% |

The authors did not publish any splits or resources for the *Visual Genome* dataset. In our study, we followed the approach outlined in the original paper: we identified image captions containing keywords mentioned in the paper and used them for dataset labeling. For unlabeled images, we randomly selected five captions, while for labeled images, we chose five captions that included the relevant keywords. Table 19 presents the poisoning ratios relative to both dataset and class size, as employed in our experiments.

Table 19: VG Poisoning rates

| Goal | Class Samples | Poisoned Samples | Class poisoning rate | Dataset poisoning rate |
|---|---|---|---|---|
| sheep2aeroplane | 1275 | 1275 | 100% | 1.18% |
| | | 864 | 67.5% | 0.8% |
| | | 540 | 42.5% | 0.5% |
| | | 108 | 8.5% | 0.1% |
| | | 86 | 6.7% | 0.08% |

