# OpenReview forum: "Re: Data Poisoning Attacks Against Multimodal Encoders"
_TMLR — Rejected by TMLR_

### Review · Reviewer_y8k6 · 2024-03-26

**Summary Of Contributions:**

The paper tests a set of existing attacks and defenses against multimodal autoencoders, showing some limitations, flaws, and additional observations with respect to the work in Yang et al. 2023. More concretely, the contributions of the paper are as follows:
+ The authors analyze the effect of the poisoning attacks considering the ratio of poisoned examples with respect to the size of the target class, not with respect to the total number of training points as in previous works. They show this is a better indicator that determines the success of the attack and that, just considering the ration of poisoning with respect to the whole training set can provide misleading results.
+ The authors perform transfer attacks across different datasets and different poisoning rates, completing the analysis in previous works.
+ The authors replicate experiments with two existing defenses showing limitations in one of them and proposing a variant for another one.

**Audience:**

Yes

**Claims And Evidence:**

Yes

**Requested Changes:**

+ Provide more background (see previous comments).
+ Include a section with the state of the art.
+ Highlight better in the paper the changes made to the attacks and defenses with respect to Yao et al.’s paper. For instance, in Section 3.2, it is unclear if the poisoning rates reported are with respect to the total number of training data points or with respect to the number of points to the target class. I think that making the differences more explicit can help the reader to follow better the empirical analysis.

**Strengths And Weaknesses:**

Strengths:
The authors strived to provide a more complete overview of the work proposed by Yang et al. 2023, showing some interesting observations that were overlooked in this previous paper (see the contributions listed before). For example, the authors show that, to have a more informative analysis of the robustness of these models to data poisoning, it is more relevant to analyze the ratio of poisoning for each target class. According to the authors, some of the results reported by Yang et al. 2023, where the ratio of poisoning was reported with respect to the total number of training data points, provide an overly pessimistic view on the robustness of these multimodal autoencoders.

Limitations:
The paper does not provide a novel contribution to the state of the art. It relies entirely on attacks and defensive algorithms already proposed in the research literature and re-used the code made available by Yang et al. 2023 to replicate the experiments. According to TMLR’s submission guidelines on dual submissions and originality, it is not clear to me if this paper contravenes TMLR’s policies. Perhaps the area chair can provide further clarifications on this. I believe the authors made an honest analysis of the limitations of the contribution by Yao et al. 2023, making good observations that can be of interest to the research community. However, it is unclear to me if this fits in the scope of TMLR.

Some comments:
+ The paper mostly focuses on the experimental work but neglects the background and the related work. For example, the background is very brief and does not provide a good understanding of the problem considered in the paper. It would be necessary to provide more context and a more detailed description of the attacks and defenses, as well as the threat model considered. In this section, it would also be necessary to cite explicitly the original source for the attacks and defenses used in the paper. Similarly, it would be good to have a section with the state of the art.
+ For the results in Table 4 the authors say: “The MinRank values indicate the minor discrepancies between our results and those of the original paper”. However, the values reported in Table 4 show quite significant differences for the MinkRank metric. Could the authors explain why? Similar comments apply to the results in Table 5, where the discrepancies seem significant too.

---

> ### Author Response · Authors · 2024-04-01
> **Response**
>
> Thank you for your valuable review.
>
> We wanted to clarify your comments here,
>
> ***
>
> ## **Novel Contribution:**
>
> This paper serves as a reproducibility study. Our primary objective is to delve deeper into and experiment with the novel approaches employed in both attacking and defending the multimodal models introduced by Yang et al. in 2023. It’s important to note that our work is not an extension of theirs; instead, we offer a comprehensive examination that sheds light on previously ambiguous or overlooked aspects. Our contribution lies in rigorously testing the generalizability and effectiveness of these attack and defense strategies.
>
> ***
>
> ## **Dual Submissions and Originality:**
>
> This work is by no means an expanded version of the original work but rather a reproducibility study of the work done by Yang et al. 2023. No figures, text or tables were taken from the original work, but all results are reproduced results using the same or different settings as mentioned in our work. We consider our work as part of “reproducibility studies of previously published results or claims;” which is within the scope TMLR’s papers.
>
> ***
>
> ## **Code Reuse:**
>
> As mentioned in section 3.1, the code provided by Yang et al. 2023 was useful but cannot be used to run all the experiments. The code was entirely missing the defenses and attack transferability in addition to some minor changes in order to run the other experiments. Our main contribution is not the code, however, once a decision is made the repo with new code containing all fixes and missing experiments will be made public. Our code will contain scripts to run each experiment separately to make our results easy to replicate.
>
> ***
>
> ## **Background and SOTA:**
> The background section provided can be extended to include the threat model and other information to help the reader. However, as this is a reproducibility study we assumed that the reader will already have read the paper by Yang et al. 2023, making it unnecessary to explain the background again. But this could be a good addition to make understanding this work independent of the work done by Yang et al. 2023. Regarding the SOTA, Yang el al. 2023 considered their work as the first to poison both modalities with no previous similar work. Therefore, no SOTA section was added following what they did.
>
> ***
>
> ## **Results and clarity:**
>
> This can be an actual issue for the reader, we will change this to make it clearer. To clarify, for table 4 and 5, the difference between the MinRank values is considered small. This is because MinRank represents the approximate appearance of the image in the rank list. For example, the MinRank in table 4 for the flickr-pascal dataset is 51.048 and 2.192, these ranks can be considered close to the 49.248 and 3.6 in our findings. We considered this a minor difference as there are 1 or 2 position differences in the rank list. In section 3.2 the poisoning rates are relative to the entire dataset. We added a table in the appendix with all the poisoning rates to prevent this confusion. Making the difference more explicit can be done to make it easier for the reader.

---

### Review · Reviewer_GSiU · 2024-03-31

**Summary Of Contributions:**

1. The paper runs a re-evaluation of the prior work [1] and shows several discrepancies from the original work's results.
2. The paper presents an improved defense against data poisoning attacks on multi-modal encoders.

[1] Yang et al., Data poisoning attacks against multimodal encoders. ICML'23

**Audience:**

Yes

**Broader Impact Concerns:**

No discussion could be found in the paper, but I don't have any ethical concerns.

**Claims And Evidence:**

Yes

**Requested Changes:**

Please refer to the detailed comments section.

**Strengths And Weaknesses:**

Strengths
1. The paper performs a reproduction study of the prior work by Yang et al.
2. The paper identifies the conservative cases where the attack's success does not match the reported numbers.

Weaknesses
1. The paper's findings and lessons are too focused on the work by Yang et al.
2. Some attack configurations do not seem to be the same as the prior work [1]
3. The novelty of the proposed defense is weak.

Detailed comments (Including requested changes)

While we encourage the re-evaluation of the claims in any previous studies, I believe the focus of the studies should be to find "generalizable lessons" for the community such that future work can follow the desirable practices. Considering this, this work unfortunately too much focuses on criticizing the results in prior work by Yang et al.

I believe that we can find many previous studies when one reproduces the results, we cannot reproduce the "exact numbers" shown in the paper. Perhaps, this is because of the stochasticity of the area we study (i.e., machine learning). So the important points we focus on are:

1. Does the fact that we don't reproduce the exact numbers hurt their novelty?
2. What does it imply for future work who want to improve the attack (by [1]) and propose defenses?

I believe, that even if the numbers are not the same, the novelty of the prior work [1] stays the same. The (victim) work shows one can compromise the training data used for multi-modal encoders to induce malicious objectives. The question perhaps brought by the authors is whether the attack is practical or not. But since the work [1] does NOT claim that they are the strongest attack, demonstrating in some cases that the attack works is fine to me.

So what about the other factor? It's a bit unfortunate to see there is already prior work discussing desirable practices of data poisoning attack evaluation [2]. This work claims that we need to run multiple attacks on multiple target samples/datasets/models to see the amortized attack success in practical scenarios, which is an exact contribution this paper aims to bring.

[2] Schwarzschild et al., Just How Toxic is Data Poisoning? A Unified Benchmark for Backdoor and Data Poisoning Attacks

In addition to that, I found that this work uses their "code" to reproduce the results while the setting could be different:

1. Does this work compromise the same training samples?
2. Does this work use the same training configuration as the authors?
3. Does this work run multiple attacks and compute the average?

Depending on the samples we compromise, the attack success could differ a few percentage points, e.g., in clean-label poisoning attacks, we see the attacks on the same target sample are successful in 7 over 10 runs. It would be nice to have a table contrasting or comparing the attack and training configurations between this work and the prior one(s) to clarify this concern.

For the new defense proposal, I believe there have been many works proposing to use a small subset of clean samples as a trust base to identify poisoning samples or make the attack ineffective aftermath. So, I believe that it would be less surprising to the community that fine-tuning a compromised model with a clean subset of data can render the attack ineffective.

Moreover, the defense evaluation unfortunately is caught in the same trap: the results are not from running over multiple targets/poisoning samples/models/hyper-parameters; thus, it may be hard to claim the effectiveness.

---

> ### Author Response · Authors · 2024-04-01
> **Response**
>
> We appreciate your insightful and comprehensive review.
>
> You’ve pointed out several areas that may require modifications. We would like to provide further clarification on some of the concerns you raised:
>
> ***
>
> ## **Scope of Work:**
> This work is considered a “reproducibility study of previously published results or claims,” which is why we heavily focus on the work by Yang et al. 2023. As mentioned in your comments, our goal was to test the generalizability of these claims. However, when trying to understand the relationship between the poisoning rate and dataset size, we discovered that the model hardly trains on any clean data from the target class. This led us to focus on the authors’ main claim that by poisoning a small portion of the dataset, we can achieve the desired behavior. We proved that this is not entirely true, as they assumed the rest of the dataset doesn’t contain data about the specific poisoned class. This might not be the case in a real-world scenario where data from an important class (important because it seems to benefit the attacker) will be chosen more accurately. Our results show that poisoning some of these class samples won’t be effective when a greater number of clean samples are present in the dataset.
>
> ***
>
> ## **Experiment Configuration:**
> For the initial experiments, we utilized the authors' code, which we then modified to implement the defenses and transferability, in addition to making some changes. We highlighted the differences between our results and those of Yang et al. 2023 in the first experiment, as we anticipated obtaining identical results using their code. Yang et al. 2023 published the splits used for all datasets except the VG dataset. The code also included all the hyperparameters and seeds used. The authors did not provide an average, but we presume they ran the experiments multiple times. We replicated this by running the experiments multiple times and reporting the results closest to what they reported. All experiments up to Table 6 and Figure 5 should exactly match the configuration used by the authors. The code we will provide will include scripts to run each experiment separately, ensuring the replicability of our results.
>
> ***
>
> ## **Effect of results on Yang et al. 2023 work:**
> The work of Yang et al. 2023 is recognized as the first to poison both modalities, lending it significant value. Our results do not deny  their novelty, but offer a more transparent evaluation of the attack’s performance. We show that the attack’s effectiveness is limited and the attacker can only provide performance guarantees under the assumption that they are aware of the target class portion size within the entire dataset. This insight should influence future research, as the attack setting, which cannot provide concrete guarantees, is unreliable. We have highlighted that the attack’s performance is primarily determined by the poisoning rate relative to the target class. Again we do not deny the novelty of the work done by Yang et al. 2023 but provide further evaluation to their methods as we consider their work important being the first to attack both modalities.
>
> ***
>
> ## **The defense variant:**
> The defense strategy we propose is not a novel method, but rather a variant designed to generalize the defense outlined in the original paper. This variant is not presented as a groundbreaking approach. As previously stated, our focus is on reproducing their work and conducting a more comprehensive evaluation of the attacks and defenses. We pay particular attention to aspects that may have been overlooked in the original study, as these could have implications for future research.

---

### Review · Reviewer_ghJd · 2024-04-05

**Summary Of Contributions:**

Yang et al. (2023) conducted a study on the susceptibility of multimodal models to such attacks, introducing three types of poisoning attacks and two potential defense strategies. This paper replicates Yang et al. (2023) paper attacks, finding their efficiency varies with the poisoning rate relative to the target class size. The authors also confirm previous findings and propose improvements to the defense strategies, highlighting the need for robust defenses against such attacks.

**Audience:**

Yes

**Claims And Evidence:**

Yes

**Requested Changes:**

See above weaknesses.

**Strengths And Weaknesses:**

**Strengths**
1. This paper is well-organized and easy to follow.
2. The paper reproduced Yang et al. (2023)'s work and has different findings.


**Weaknesses**
1. The paper's novelty is limited and appears to be incremental based on Yang et al. (2023)'s work.
2. The code link provided in the paper abstract is unavailable, hindering reproducibility and further research.
3. It's unclear why the poisoning rate for the Flickr-PASCAL and COCO datasets differs, potentially impacting the generalization of the findings.
4. There's no exploration of poisoning images to generate target text, leaving a gap in understanding potential attack vectors.
5. “In COCO, doubling the poisoning rate nearly doubles the attack performance, a phenomenon not observed in Flickr-PASCAL.” Could the authors explain the reason?
6. The paper does not investigate the resilience against existing state-of-the-art defense methods, such as adversarial training, leaving the effectiveness of these defenses unexplored.

---

> ### Author Response · Authors · 2024-04-05
> **Response**
>
> Thank you for taking the time to review our work. We appreciate your valuable feedback and would like to address some of the concerns you raised.
>
> ***
>
> ## **Novel Contribution:**
> This effort is a “reproducibility study of previously published results or claims” which focuses on reproducing Yang et al. (2023)'s work and testing the generalizability of the attacks proposed and effectiveness of the defenses. We did this by replicating the attacks and finding out that the poisoning rate relative to class size is the main dependency in the attacks’ performance. Unfortunately, this rate cannot be guaranteed by the attacker, which weakens the overall attack. Additionally, we thoroughly examined the defenses. While the first defense proved challenging to employ, we found that results with learning rates were unreliable due to low model utility. As part of our contribution, we also proposed a more efficient variant of the second defense.
>
> ***
>
> ## **Code:**
> The code link has been temporarily removed for blinded review. Once a decision is made, we will provide the code, which includes scripts for running experiments independently. This will allow others to replicate our results for future research.
>
> ***
>
> ## **Double poisoning rate for COCO:**
> We observed this phenomenon in the original paper but noticed that it was not adequately addressed. It may be worthwhile to investigate whether this effect is class-dependent or specific to attack three.
>
> ***
>
> ## **Experiment Settings:**
> Our experimental setup closely follows the original paper, as our work aims to reproduce their attacks and defenses. Specifically, we use the same task—image retrieval—as our evaluation metric. For replicating the attacks relative to dataset size, we applied identical poisoning rates to both COCO and Flickr-pascal datasets. Yang et al. (2023)'s work represents the first attempt at poisoning both modalities, and there are no existing state-of-the-art attacks or defenses to compare against. The original paper also did not include such comparisons.

---

### Decision · Action_Editor_6KJn · 2024-05-29

**Recommendation:** Reject

**Comment:**

Reviewers found this paper's observations about poisoning attacks and defenses interesting, but not groundbreaking. They liked how it builds on existing work and offers a better way to think about the attack's effectiveness. However, they also think it's a bit too focused on existing research and doesn't really offer insight that would be interesting to a wide audience. Further there were concerns about reproducibility of specific results and the accuracy of MinRank values.

**Audience:**

There were concerns about how relevant the proposed defense is to modern day systems.

**Claims And Evidence:**

Generally well-supported, but some concerns about reproducibility of specific results and the accuracy of MinRank values.